# Assessment of Energy Budgeting and Its Indicator for Sustainable Nutrient and Weed Management in a Rice-Maize-Green Gram Cropping System

**Dibakar Ghosh** [1,2,*,†], **Koushik Brahmachari** [2], **Anupam Das** [3,†], **Mohamed M. Hassan** [4,*], **Pijush Kanti Mukherjee** [5], **Sukamal Sarkar** [6], **Nirmal Kumar Dinda** [7], **Biswajit Pramanick** [8], **Debojyoti Moulick** [9], **Sagar Maitra** [10] **and Akbar Hossain** [11,*]

1. ICAR-Indian Institute of Water Management, Bhubaneswar 751023, Odisha, India
2. Department of Agronomy, Bidhan Chandra KrishiViswavidyalaya, Mohanpur, Nadia 741252, West Bengal, India; brahmacharis@gmail.com
3. Department of Soil Science and Agricultural Chemistry, Bihar Agricultural University, Sabour, Bhagalpur 813210, Bihar, India; anusoil22@gmail.com
4. Department of Biology, College of Science, Taif University, P.O. Box 11099, Taif 21944, Saudi Arabia
5. ICAR-Directorate of Weed Research, Jabalpur 482004, Madhya Pradesh, India; pkm_agronomy@yahoo.co.in
6. Office of the Assistant Director of Agriculture, Bhagawangola-II Block, Directorate of Agriculture, Government of West Bengal, Murshidabad 742135, West Bengal, India; sukamalsarkarc@yahoo.com
7. Office of the Assistant Director of Agriculture, Suri II Block, Department of Agriculture, Government of West Bengal, Purandapur, Birbhum 731129, West Bengal, India; nirmaldinda@gmail.com
8. Dr. Rajendra Prasad Central Agricultural University, Pusa, Samastipur 848125, Bihar, India; bipra.its4u@gmail.com
9. Plant Stress Biology and Metabolomics Laboratory, Assam University, Silchar 788011, Assam, India; drubha31@gmail.com
10. Department of Agronomy and Agroforestry, Centurion University of Technology and Management, Paralakhemundi 761211, India; sagar.maitra@cutm.ac.in
11. Bangladesh Wheat and Maize Research Institute, Dinajpur 5200, Bangladesh
* Correspondence: dibakar.ghosh@icar.gov.in (D.G.); m.khyate@tu.edu.sa (M.M.H.); akbarhossainwrc@gmail.com (A.H.)
† Authors sharing equal credit.

**Abstract:** Sustainability and climate change are the two major challenges to the agricultural production system. The trade-off between them is essential for higher profitability. The energy assessment is essential for judging the sustainability and vulnerability of a production system. Besides, nutrient management and weed management are equally imperative to sustainability. Thus, the present study was executed to assess the energy balance, key energy indicators and profitability of rice–maize–green gram system under different nutrient and weed management practices. Application of *Brassicaceous* seed meal (BSM) along with mineral fertilizer attributed the highest rice (5.62 t ha$^{-1}$) and maize (6.48 t ha$^{-1}$) yield which was 11.6%, 8.3% and 3.7% in maize and 10.0%, 6.2% and 8.7% in rice for the conjoint application with vermicompost, farmyard manure (FYM) and neem cake, respectively. Moreover, BSM recorded the highest net energy gain, energy use efficiency and energy efficiency ratio and the lowest specific energy in all the crops. Application of pre-emergence herbicides followed by hoeing was found to be best in all respects including yield, profitability, energy use efficiency, energy effectiveness, etc. The appropriate combination of integrated nutrient management with BSM and pre-emergence herbicide application followed by hoeing provided an additional advantage not only in terms of yield but also an efficient use of energy, profitability and environmental safety. BSM and neem cake could be the alternative organic manure in the integrated nutrient-cum-weed management module and they could be able to compensate the paucity of FYM and vermicompost in the country.

**Keywords:** *Brassicaceous* seed meal; energy; profitability; rice; maize; green gram

## 1. Introduction

In the Indian subcontinent, rice–wheat and rice–rice are the major rice-based cropping systems. However, the production potential of these systems has become fatigued and is plateauing [1,2], and income has started to show a declining trend [3], whereas, rice–potato, rice–mustard and rice–pulses systems are quite vulnerable under changing climatic scenarios [4]. Among these cropping systems, rice–maize could be an option to address the challenges that have emerged due to the vulnerability of the agro-ecosystem under the climate change scenario. Overdependency on high-input-driven staple cereal crops compelled us to go for widespread cultivation of cereals, resulting in the deterioration of ecological parameters, most profoundly soil parameters. Realizing the inherent benefit of legumes, notably on the soil parameters of agro-ecosystems, green gram is always a better choice for farmers in terms of receiving additional economic benefits.

For sustaining system productivity, proper management of soil nutrients is one of the major important aspects of crop cultivation. Greater nutrient availability could augment higher production. However, this concept was misleading after a few decades of Green revolution and disproportionate use of fertilizer has turned out as a potential offset for production and other ecosystem services [5]. Among them, deterioration of soil parameters may become one of the keys constraints for poor production and food insecurity [6]. Hence, the overall improvement of soil parameters needs special heed to maximize the function of the ecosystem which ultimately provides the basis of enhanced productivity. Therefore, integrated nutrient management (INM) encompassing the integration of different sources of plant nutrients has registered a key role in augmenting ecosystem services while correcting the deficiencies of soil parameters [7,8]. Use of farmyard manure (FYM), vermicompost, crop residues and green manure in integrated mode objectifying the benefits of INM has been studied in detail in the rice–maize system. However, the use of *Brassicaceous* seed meal (BSM) and neem cake is very rare as supplementation with mineral fertilizer. Moreover, the use of neem cake has been reported by a few researchers; however, those studies were mostly related to nitrification inhibitors, greenhouse gas emissions, etc. [9]. The use of BSM and neem cake forINM for suppressing weeds in a rice–maize–green gram cropping system has not been explored yet.

Among the crop challenges, weeds pose a tremendous threat to the agro-ecosystem and offset the potential benefits of the genetic makeup of the crops [10,11]. Heterogeneity of weeds in agro-ecosystems is primarily influenced by agronomic management practices, apart from fluctuating weather conditions [12]. In particular, fertilization alters soil fertility, affecting not only crop growth but also diversity and growth of associated weeds [13,14]. In the present scenario, weed management through herbicides is a widely accepted method due to the ease of application, provision of quick results and long-lasting phytotoxic effect on weeds [15–17]. However, it has already raised several issues in the backdrop of herbicide-resistant weeds due to the repeated application of herbicides having the same mode of action, less selectivity due to overdose, lack of proper understanding about herbicides, herbicide load in food chain, etc. Moreover, it becomes a trade-off between workload, quality of produce and ecological consequences. These issues emphasized the need to look for an alternative eco-friendly weed management strategy in crop production [18]. Therefore, the integration of mechanical methods of weed management with a chemical approach generally provides higher weed control efficiency than a solely chemical approach [19,20].

In modern agriculture, optimizing and increasing input energy enhances agricultural productivity [21,22]. In the present changing scenario, appropriate energy utilization is an integral part of higher productivity, profitability and sustainability of an agricultural system. It helps in conserving natural resources and in reducing the environmental pollution in addition to higher profitability [23]. An energy assessment can amalgamate the information of a system for better adoption by farmers as well as better decision making by policymakers [24]. Energetics of integrated nutrients and weed management practices is an emerging issue due to its relationship with the economics of that system [25]. No

such studies have been conducted to judge the energy utilization efficacy of varied weed and nutrient management practices in a rice–maize–green gram cropping system. While considering the immensely important attribute of the sustainability of an agro-ecosystem, the current study has been planned and executed to assess the energy balance, key energy indicators and profitability of the rice–maize–green gram system.

## 2. Materials andMethods

### 2.1. ExperimentalSite

This study was carried out at a farmer's field inMuratipur, Nadia, West Bengal, India, the research farm, BCKV, inWest Bengal, India, during two consecutive years, i.e., 2014–2016. The experimental site was situated at 88°27′ N latitude and 22°59′ E longitude, with an altitude of 7.9 m above the mean sea level. The experimental soil was typical of Gangetic alluvium (Entisol), having moderate fertility, and the initial physicochemical properties of the experimental soil are described in a previously published paper in *Agronomy* [26]. The field had an irrigation facility throughout the year and slope of the land was medium. The climate of the area is sub-tropical humid in nature and details of it are described in a previously published paper in *Agronomy* [26].

### 2.2. ExperimentalTreatments

The experiment was laid out in a randomized block design, having two factors,viz. nutrient management (5 levels) and weed management (3 levels), replicated thrice. The five levels of nutrient management practices were 100% recommended dose of nitrogen ($RD_N$) through chemical fertilizer ($NM_1$) and 25% $RD_N$ through vermicompost ($NM_2$), FYM ($NM_3$), BSM ($NM_4$) and neem cake ($NM_5$) (Table S1). The remaining amount of $RD_N$ (75%) for $NM_2$ to $NM_4$ and a full dose of phosphorus and potassium were applied through the chemical fertilizers urea, single super-phosphate and muriate of potash, respectively. The recommended dose of fertilizer (RDF) for rice and maize was 200–60–60 and 60–30–30 kg N–$P_2O_5$–$K_2O$ ha$^{-1}$, respectively [27]. The full doses of organic manures, phosphatic and potassic fertilizers were applied before final land preparation, whereas N fertilizer was applied in three doses in both the crops. The nutrients were applied in rice and maize crops, whereas green gram was grown under residual soil fertility. The three levels of weed management practices are presented in Table 1. Herbicides were applied with a knapsack sprayer of 16-liter capacity with a flat fan nozzle and the spray volume was 500 L ha$^{-1}$. For mechanical weeding, a wheel hoe was used to remove the weeds in between the rows of crops, and in the weedy control treatment, the weeds were left uncontrolled.

**Table 1.** Weed management practices in different crops under the rice–maize–green gram system.

| Treatments | Rice | Maize | Green Gram |
|---|---|---|---|
| WM$_1$-Control | Unweeded | Unweeded | Unweeded |
| WM$_2$-Herbicidal | Bispyribac-sodium 25 g/ha at 15 days after transplanting (DAT) followed by metsulfuron-methyl + Chlorimuron ethyl (2 + 2) g/ha at 30 DAT | Atrazine 1000 g/ha as pre-emergence at 2 days after sowing (DAS) | Imazethapyr 100 g/ha at 25 DAS |
| WM$_3$-Integrated | Bensulfuron methyl + Pretilachlor (60 + 600) g/ha at 5 DAT followed by hoeing at 30 DAT | Atrazine 1000 g/ha at 2 DAS followed by hoeing at 30 DAS | Pendimethalin 750 g/ha at 2 DAS followed by hoeing at 25 DAS |

### 2.3. Crop Management

The rice crop (cv. *Satabdi* (IET 4786)) was transplanted manually with the spacing of 20 cm row-to-row and 15 cm plant-to-plant; however, maize (cv. P-3396) and green gram(cv. *Samrat* (PDM 139))were sown with the row spacing of 60 and 30 cm, respectively. The plant-to-plant spacing of maize and green gram was maintained at 30 and 5 cm, respectively. In

rice and maize, for controlling stem borer, fipronil 5% Suspension Concentrate was used, whereas spinosad 45% SC was applied to control green gram pod borer. The gross and net plot sizes were 7.2 × 3.0m and 6.0 × 2.0m, respectively. The net plot area was used for the determination of crop yields. The crops were harvested manually at physiological maturity and yield was taken at 14% moisture level.

### 2.4. Method of Energ Calculation and Indicators

In Table 2 and Tables S2–S7.

**Table 2.** Energy input and output of individual crop was calculated using an energy co-efficient for each treatment.

| Component | | Unit | Energy Equivalent Coefficient (MJ/Unit) | Remarks |
|---|---|---|---|---|
| A. | Inputs | | | |
| a. | Human labour | | | |
| 1. | Adult man | Man-hour | 1.96 | |
| 2. | Woman | Woman-hour | 1.57 | 1 adult woman = 0.8 adult man |
| b. | Animals | | | |
| Bullocks (Medium) | | Pair-hour | 10.10 | Body weight, 352-450 kg |
| c. | Diesel | Liter | 56.31 | It includes the cost of lubricant |
| d. | Farm machinery | kg | 62.7 | |
| e. | Chemical Fertilizer | | | |
| 1. | N | kg | 60.6 | |
| 2. | $P_2O_5$ | kg | 11.1 | |
| 3. | $K_2O$ | kg | 6.7 | |
| f. | Organic manure | | | |
| Vermicompost/ Farmyard manure/ *Brasecacious* seed meal/ Neem cake | | kg (dry mass) | 0.3 | |
| g. | Chemicals | | | |
| 1. | Herbicide | kg | 254.45 | Chemical requiring dilution at the time of application |
| 2. | Pesticide | kg | 184.63 | |
| h. | Seed | | | |
| 1. | Maize | Kg | 14.7 | Same as that of output of crop production system |
| 2. | Green gram | Kg | 14.7 | Same as that of other pulse crops |
| 3. | Rice | kg | 14.7 | Same as that of the output of crop production |
| B. | Outputs | | | |
| a. | Main product | | | |
| 1. | Maize | kg (dry mass) | 14.7 | The main output is grain |
| 2. | Green gram | kg (dry mass) | 14.7 | The main output is seed |
| 3. | Rice | kg (dry mass) | 14.7 | The main output is grain |
| b. | By product | | | |
| Stover/straw | | kg (dry mass) | 12.5 | |

Source: Mittal et al. [28]; Mittal and Dhawan [29]; Singh et al. [30]; Parihar at al. [31].

Energy input and output of individual crops were calculated using energy coefficients for all the treatments [32,33].

Energy indicators were calculated using the following equations [34,35]:

$$\text{Net Energy} = \text{Energy output (MJ ha}^{-1}) - \text{Energy input (MJ ha}^{-1}) \tag{1}$$

$$\text{Energy Use Efficiency} = \frac{\text{Total Output Energy} \left(\text{MJ ha}^{-1}\right)}{\text{Total Input Energy} \left(\text{MJ ha}^{-1}\right)} \tag{2}$$

$$\text{Specific Energy}\left(\text{MJ kg}^{-1}\right) = \frac{\text{Total Intput Energy}\left(\text{MJ ha}^{-1}\right)}{\text{Total Main Product Yield}\left(\text{kg ha}^{-1}\right)} \tag{3}$$

$$\text{Energy Intensiveness}\left(\text{MJ Rs.}^{-1}\right) = \frac{\text{Total Intput Energy}\left(\text{MJ ha}^{-1}\right)}{\text{Cost of Cultivation}\left(\text{Rs. ha}^{-1}\right)} \tag{4}$$

$$\text{Energy Efficiency Ratio} = \frac{\text{Total Output Energy in Main Product}\left(\text{MJ ha}^{-1}\right)}{\text{Total Intput Energy}\left(\text{MJ ha}^{-1}\right)} \tag{5}$$

### 2.5. Statistical Analysis

The statistical analysis of data was performed using SAS 9.3. The data were subjected to ANOVA. The year effect was significant and data were depicted yearly. Treatment means were separated using Tukey's Honestly Significant Difference test for post-hoc analysis at the 5% level of significance. Correlations between parameters were computed where applicable. The benefit: cost ratio (B:C ratio) was calculated according to Babu et al. [36]. Excel software (version 2016, Microsoft Inc., Washington, DC, USA) was used to draw graphs and figures.

## 3. Results and Discussion

### 3.1. Crop Productivity

Maize and rice grain yields were significantly improved by the conjoint application of organics (vermicompost, FYM, BSM and neem cake) with mineral fertilizer application than sole mineral fertilization (NM$_1$); however, green gram seed yield was showed albeit non-significance with nutrient management (Table 3). Besides nutrient management, weed management has a profound influence on grain yield in all crops. The highest grain yield of maize (6.48 t ha$^{-1}$) and rice (5.62 t ha$^{-1}$) was obtained in BSM-applied plots. Application of vermicompost, FYM and neem cake resulted in equal yields of all the three crops. The superiority of BSM was attributed in terms of higher yield of 13.7%, 11.6%, 8.3% and 3.7% in maize and 11.9%, 10.0%, 6.2% and 8.7% in rice over sole mineral fertilizer (NM$_1$) and conjoint application with vermicompost (NM$_2$), FYM (NM$_3$) and neem cake (NM$_5$), respectively. A slight increase in greengram yield (1.8–4.6%) was recorded under the neem cake application over other treatments. Application of pre-emergence herbicides followed by hoeing realized the highest yield in each crop and ascribed 8.2–32.9%, 10.9–23.7% and 9.6–25.8% higher yields in maize, rice and green gram, respectively, over other weed management practices.

### 3.2. Energy Utilization by Crops

The energy utilization in the cropping system was very much crop-specific and strongly aligned with the management practices. The energy utilized by the crops during their growth period is depicted in Table 4. The total energy utilized by the crops has been estimated from the energy derived from land preparation, seed, plant protection and surveillance, labour and diesel. The relative distribution of energy in different sources was the highest in diesel (59.62–77.61%), followed by human labour (17.32–26.70%), seed (2.43–9.55%), land preparation (0.77–2.82%) and irrigation (0.03–0.05%). Rice crops utilized a considerable amount of energy (8.34%) during fertilization in the nursery bed. The highest amount of energy was utilized by the maize crop (9069.26 MJ ha$^{-1}$) followed by rice (7697.18 MJ ha$^{-1}$) and green gram (5050.67 MJ ha$^{-1}$).

**Table 3.** Grain yield of different crops influenced by management practices (mean of two years) under the rice–maize–green gram cropping system.

| Treatments | Grain Yield (t ha$^{-1}$) | | |
| --- | --- | --- | --- |
| | Rice | Maize | Green Gram |
| Nutrient management practices | | | |
| NM$_1$ | 4.95 [b] | 5.59 [*,c] | 0.746 [a] |
| NM$_2$ | 5.06 [b] | 5.73 [b,c] | 0.747 [a] |
| NM$_3$ | 5.27 [a,b] | 5.94 [a,b,c] | 0.769 [a] |
| NM$_4$ | 5.62 [a] | 6.48 [a] | 0.770 [a] |
| NM$_5$ | 5.13 [b] | 6.24 [a,b] | 0.785 [a] |
| Weed management practices | | | |
| WM$_1$ | 4.48 [C] | 4.66 [C] | 0.642 [C] |
| WM$_2$ | 5.23 [B] | 6.38 [B] | 0.782 [B] |
| WM$_3$ | 5.87 [A] | 6.95 [A] | 0.865 [A] |
| *p*-value | | | |
| NM | 0.0006 | 0.0001 | 0.2750 |
| WM | <0.0001 | <0.0001 | <0.0001 |
| NM × WM | 0.7281 | 0.1619 | 0.6079 |

Treatment details are described in Table 1; * values followed by the same letter are not significantly different at $p < 0.05$. Effects of nutrient source as lower case and effect of weed management as upper case.

**Table 4.** The sources utilization of energy (in MJ ha$^{-1}$) for raising various crops under the rice–maize–greengram cropping system.

| Field Operation | Rice | % * | Maize | % * | Green Gram | % * |
| --- | --- | --- | --- | --- | --- | --- |
| Land preparation | 216.68 | 2.82 | 110.02 | 1.21 | 38.88 | 0.77 |
| Seed | 735.00 | 9.55 | 220.50 | 2.43 | 367.50 | 7.28 |
| Irrigation | 2.59 | 0.03 | 4.62 | 0.05 | 1.85 | 0.04 |
| Fertilizer in nursery | 641.60 | 8.34 | - | - | - | - |
| Plant protection and surveillance | 124.92 | 1.62 | 124.92 | 1.38 | 27.98 | 0.55 |
| Labour | 1387.08 | 18.02 | 1570.45 | 17.32 | 1348.48 | 26.70 |
| Diesel | 4589.31 | 59.62 | 7038.75 | 77.61 | 3265.98 | 64.66 |
| Renewable energy [†] | 2124.67 | - | 1795.57 | - | 1717.83 | - |
| Non-renewable energy [‡] | 5572.51 | - | 7273.69 | - | 3332.84 | - |

* Represents the % value of each input source relative to the total input energy; [†] renewable energy sources include seed, irrigation and human labour and [‡] non-renewable energy sources were land preparation, diesel, fertilizer and plant protection chemicals.

### 3.3. Influence of Management Practices on Energy Indicators in Different Crops

Energy indicators such as net energy gain, energy use efficiency, specific energy, energy intensiveness and energy efficiency ratio were used to discriminate the efficiency of a crop or cropping system in utilizing energy under a set of management practices. Every indicator has a special aptitude to reveal efficacy. The net energy gain is simply a balance sheet between output and input energies. Energy use efficiency is used to express the effectiveness of an agricultural production system, whereas energy efficiency ratio considers the main product as a contributor to the pool of output energy [34,37]. Specific energy is an estimated yield as a function of input energy, where a higher value indicates lower efficiency of crop or cropping system.

3.3.1. Net Energy Gain

The nutrient management practices did not show any significant ($p < 0.05$) variation in net energy gain, except the rice crop (Tables 5–7). However, N supplementation through BSM (NM$_4$) recorded the highest net energy gain in all the crops among the different nutrient management practices. Weed management practices had significant differences in net energy gain in all the crops, where herbicide application followed by hoeing recorded

the highest net energy gain. Interaction of nutrient management and weed management practices did not result in any significant variation in net energy gain. The higher net energy gain may be due to a higher yield from the respective management practices [34,38,39].

**Table 5.** Energy indicators in maize under the rice–maize–green gram cropping system.

| Treatments | Net Energy Gain (MJ) | | Energy Use Efficiency | | Specific Energy (MJ/kg) | | Energy Intensiveness (MJ Rs$^{-1}$) | | Energy Efficiency Ratio | |
|---|---|---|---|---|---|---|---|---|---|---|
| | Year I | Year II | Year I | Year II | Year I | Year II | Year I | Year II | Year I | Year II |
| **Nutrient management practices** | | | | | | | | | | |
| NM$_1$ | 155,059 *[a] | 157,632 [a] | 7.78 [b] | 7.90 [b] | 4.26 [a] | 4.16 [a] | 3.41 [a] | 3.45 [a] | 3.55 [b] | 3.64 [c] |
| NM$_2$ | 151,933 [a] | 162,175 [a] | 8.43 [a,b] | 8.92 [a,b] | 3.88 [a,b] | 3.69 [a] | 3.05 [a,b] | 3.23 [a,b] | 3.99 [a,b] | 4.23 [b] |
| NM$_3$ | 156,064 [a] | 167,040 [a] | 8.63 [a,b] | 9.18 [a] | 3.75 [a,b] | 3.32 [b] | 3.12 [a,b] | 3.32 [a] | 4.04 [a,b] | 4.51 [a,b] |
| NM$_4$ | 166,889 [a] | 177,461 [a] | 9.29 [a] | 9.81 [a] | 3.34 [b] | 3.11 [b] | 2.77 [b,c] | 2.94 [b] | 4.55 [a] | 4.90 [a] |
| NM$_5$ | 156,135 [a] | 169,727 [a] | 8.76 [a,b] | 9.44 [a] | 3.50 [b] | 3.16 [b] | 2.33 [c] | 2.51 [c] | 4.35 [a] | 4.76 [a,b] |
| **Weed management practices** | | | | | | | | | | |
| WM$_1$ | 131,784 [C] | 140,660 [C] | 7.52 [B] | 7.95 [B] | 4.63 [A] | 4.30 [A] | 2.60 [B] | 2.77 [B] | 3.26 [C] | 3.52 [B] |
| WM$_2$ | 160,274 [B] | 171,588 [B] | 8.71 [A] | 9.23 [A] | 3.46 [B] | 3.15 [B] | 3.05 [A] | 3.22 [A] | 4.29 [B] | 4.75 [A] |
| WM$_3$ | 179,590 [A] | 188,173 [A] | 9.50 [A] | 9.91 [A] | 3.14 [B] | 3.02 [B] | 3.15 [A] | 3.28 [A] | 4.74 [A] | 4.95 [A] |
| ***p*-value** | | | | | | | | | | |
| NM | 0.5980 | 0.1402 | 0.0542 | 0.0004 | 0.0062 | 0.0001 | <0.0001 | <0.0001 | 0.0003 | <0.0001 |
| WM | <0.0001 | <0.0001 | <0.0001 | <0.0001 | <0.0001 | <0.0001 | 0.0002 | <0.0001 | <0.0001 | <0.0001 |
| NM × WM | 0.3425 | 0.2157 | 0.3617 | 0.2456 | 0.8920 | 0.4167 | 0.3820 | 0.2032 | 0.4503 | 0.3860 |

Treatment details are described in Table 1; * Values followed by the same letter were not significantly different at $p < 0.05$. Effects of source of nutrient as lower case and effect of weed management as upper case.

**Table 6.** Energy indicators in rice under the rice–maize–green gram cropping system.

| Treatments | Net Energy Gain (MJ) | | Energy Use Efficiency | | Specific Energy (MJ/kg) | | Energy Intensiveness (MJ Rs$^{-1}$) | | Energy Efficiency Ratio | |
|---|---|---|---|---|---|---|---|---|---|---|
| | Year I | Year II | Year I | Year II | Year I | Year II | Year I | Year II | Year I | Year II |
| **Nutrient management practices** | | | | | | | | | | |
| NM$_1$ | 136,566 *[a] | 138,071 [b] | 11.59 [b] | 11.70 [c] | 2.62 [a] | 2.67 [a] | 3.46 [a] | 3.49 [a] | 5.66 [b] | 5.57 [b] |
| NM$_2$ | 142,722 [a,b] | 142,054 [a,b] | 12.69 [a] | 12.61 [b,c] | 2.40 [a,b] | 2.48 [a,b] | 3.46 [a] | 3.44 [a] | 6.13 [a,b] | 6.02 [a,b] |
| NM$_3$ | 143,089 [a,b] | 147,926 [a,b] | 12.68 [a] | 13.07 [a,b] | 2.41 [a,b] | 2.35 [a,b] | 3.47 [a] | 3.57 [a] | 6.19 [a,b] | 6.41 [a] |
| NM$_4$ | 153,678 [a] | 156,005 [a] | 13.69 [a] | 13.83 [a] | 2.19 [b] | 2.22 [b] | 3.46 [a] | 3.50 [a] | 6.78 [a] | 6.65 [a] |
| NM$_5$ | 142,510 [a,b] | 145,008 [a,b] | 12.71 [a] | 12.92 [a,b] | 2.40 [a,b] | 2.38 [a,b] | 3.05 [b] | 3.10 [b] | 6.16 [a,b] | 6.21 [a,b] |
| **Weed management practices** | | | | | | | | | | |
| WM$_1$ | 129,714 [C] | 128,074 [C] | 12.44 [B] | 12.30 [B] | 2.48 [A] | 2.63 [A] | 3.35 [A] | 3.30 [A] | 5.96 [B] | 5.66 [B] |
| WM$_2$ | 144,904 [B] | 147,529 [B] | 13.68 [A] | 13.90 [A] | 2.22 [B] | 2.19 [B] | 3.40 [A] | 6.46 [A] | 6.68 [A] | 6.77 [A] |
| WM$_3$ | 156,522 [A] | 161,836 [A] | 11.90 [A] | 12.27 [B] | 2.51 [A] | 2.45 [A] | 3.40 [A] | 3.50 [A] | 5.92 [B] | 6.09 [B] |
| ***p*-value** | | | | | | | | | | |
| NM | 0.0130 | 0.0182 | 0.0001 | 0.0005 | 0.0095 | 0.0051 | 0.0004 | 0.0021 | 0.0096 | 0.0075 |
| WM | <0.0001 | <0.0001 | <0.0001 | <0.0001 | 0.0024 | <0.0001 | 0.7403 | 0.0700 | 0.0017 | <0.0001 |
| NM × WM | 0.5585 | 0.3389 | 0.6542 | 0.4556 | 0.8955 | 0.6614 | 0.4574 | 0.3209 | 0.8010 | 0.6702 |

Treatment details are described in Table 1; * Values followed by the same letter were not significantly different at $p < 0.05$. Effect of source of nutrient is lower case and effect of weed management is upper case.

**Table 7.** Energy indicator in green gram under rice–maize–green gram cropping system.

| Treatments | Net Energy Gain (MJ) | | Energy Use Efficiency | | Specific Energy (MJ/kg) | | Energy Intensiveness (MJ Rs$^{-1}$) | | Energy Efficiency Ratio | |
|---|---|---|---|---|---|---|---|---|---|---|
| | Year I | Year II | Year I | Year II | Year I | Year II | Year I | Year II | Year I | Year II |
| | *Nutrient management practices* | | | | | | | | | |
| NM$_1$ | 41,329 * [a] | 42,880 [a] | 8.40 [a] | 8.68 [a] | 7.63 [a] | 7.48 [a] | 1.32 [a] | 1.36 [a] | 1.94 [a] | 1.99 [a] |
| NM$_2$ | 41,496 [a] | 44,361 [a] | 8.43 [a] | 8.98 [a] | 7.77 [a] | 7.32 [a] | 1.32 [a] | 1.41 [a] | 1.91 [a] | 3.02 [a] |
| NM$_3$ | 40,627 [a] | 43,774 [a] | 8.24 [a] | 8.58 [a] | 7.70 [a] | 6.99 [a] | 1.29 [a] | 1.39 [a] | 1.92 [a] | 2.12 [a] |
| NM$_4$ | 41,912 [a] | 44,892 [a] | 8.49 [a] | 9.02 [a] | 7.52 [a] | 7.16 [a] | 1.33 [a] | 1.42 [a] | 1.96 [a] | 2.07 [a] |
| NM$_5$ | 40,662 [a] | 43,880 [a] | 8.31 [a] | 8.85 [a] | 7.48 [a] | 6.95 [a] | 1.30 [a] | 1.39 [a] | 1.98 [a] | 2.13 [a] |
| | *Weed management practices* | | | | | | | | | |
| WM$_1$ | 35,484 [B] | 39,377 [C] | 8.03 [B] | 8.80 [B] | 8.14 [A] | 7.65 [A] | 1.25 [B] | 1.37 [A] | 1.81 [C] | 1.93 [B] |
| WM$_2$ | 42,757 [A] | 44,727 [B] | 9.01 [A] | 9.38 [A] | 7.12 [B] | 6.60 [B] | 1.38 [A] | 1.44 [A] | 2.07 [A] | 2.24 [A] |
| WM$_3$ | 45,375 [A] | 47,769 [A] | 8.09 [A] | 8.46 [B] | 7.61 [B] | 7.28 [A] | 1.31 [A,B] | 1.37 [A] | 1.94 [B] | 2.03 [B] |
| | *p*-value | | | | | | | | | |
| NM | 0.9222 | 0.7366 | 0.9300 | 0.7225 | 0.7489 | 0.0930 | 0.9214 | 0.7490 | 0.7843 | 0.1211 |
| WM | <0.0001 | <0.0001 | 0.0003 | 0.0004 | <0.0001 | <0.0001 | 0.0031 | 0.0488 | <0.0001 | <0.0001 |
| NM × WM | 0.2840 | 0.3866 | 0.3442 | 0.4144 | 0.6087 | 0.3165 | 0.3276 | 0.4234 | 0.5450 | 0.3555 |

Treatment details are described in Table 1; * Values followed by the same letter were not significantly different at *p* < 0.05. Effect of source of nutrient islower case and effect of weed management isupper case.

### 3.3.2. Energy Use Efficiency

Energy use efficiency (EUE) was significantly varied with the crops as well as the management practices (Tables 5–7). The highest EUE was found in NM$_4$ in all the crops (maize—9.55; rice—13.76; green gram—8.76) (mean of two years). Importantly, conjoint application of organic manure and a mineral fertilizer resulted in a higher EUE than sole mineral fertilizer application. Thus, the EUE values in NM$_2$, NM$_3$ and NM$_5$ were at par. This could be due to the beneficial aspect of organic manure application resulting in higher yield over sole mineral fertilizer application [6,7]. On the other hand, the EUE values were increased to the range of 1.54–7.76% in maize, 0.95–1.65% in rice and 3.33–6.50% in green gram in the second year over the first year. However, the increment in EUE was more pronounced in the treatment where organic manure was supplemented with nitrogen dose from mineral fertilizer.

### 3.3.3. Specific Energy, Energy Intensiveness and Energy Efficiency Ratio

Specific energy (SE) was significantly varied with the management practices in maize and rice crops (Tables 5–7). Nutrient addition through BSM resulted in the lowest SE (3.23) among all the nutrient management practices (3.23–4.21). It recorded 23.3% less SE than the sole chemical fertilizer treatment (4.21) in maize. Among the organics, BSM use resulted in 17.2%, 9.4% and 3.1% lesser SE than vermicompost, FYM and neem cake, respectively, in maize. A similar trend was recorded in rice, where NM$_4$ (2.21) reduced the SE by 19.7%, 10.4%, 7.7% and 8.1% as compared to NM$_1$ (2.65), NM$_2$ (2.44), NM$_3$ (2.38) and NM$_5$ (2.39), respectively.

Energy intensiveness implies the cost of energy involved in a particular treatment. Treatment variability in energy intensiveness was more pronounced in maize rather than rice and green gram. Energy intensiveness was the lowest in neem cake-applied plots in all the crops (Tables 5–7). Energy efficiency ratio (EER) denotes how efficiently the main produce can exploit input energy. In the experiment, the energy efficiency ratio followed the same trends as energy use efficiency. N supplementation by BSM resulted in the highest EER (4.73) among all the nutrient management practices, and that was 24% and 3.7–13.1% higher than 100% NPK applied through mineral fertilizer (NM$_1$—3.60) and conjoint application of mineral fertilizer and organics (vermicompost, FYM and neem cake),

respectively, in maize. Similar trends were registered in rice and green gram; however, green gram showed a non-significant effect at $p = 0.05$.

The Pearson's correlation study showed that energy output was highly ($p < 0.01$) correlated with the cost of cultivation ($r = 0.929$, $p < 0.01$), net return ($r = 1.0$, $p < 0.01$), B:C ratio ($r = 0.669$, $p < 0.01$) and economic efficiency ($r = 0.917$, $p < 0.01$) (Tables 8 and 9). It is clear from Figure 1 that energy output was linked with energy input ($y = 1.180x + 17.727$, $R^2 = 0.502$, $p = 0.005$). Thus, the concordance between energy flow and economics helps to judge best management practices under a specified system. It was apparent that the application of a pre-emergence herbicide followed by hoeing (WM$_3$) was found to be the best among the weed management practices in all respects such as yield, profitability, energy use efficiency, energy effectiveness, etc. Similarly, the conjoint application of mineral fertilizer and BSM (NM$_4$) attributed superiority over other nutrient management practices. The appropriate combination of NM$_4$ and WM$_3$ provided additional advantages not only in terms of yield but also an efficient use of energy. Although the interaction of NM$_4$ and WM$_3$ consumed 4.49% more energy in the system, it returned the highest energy output of 38.18% in terms of additional yield (Table 10). Similarly, application of neem cake instead of BSM utilized 4.43% higher energy input and returned 26.57% energy as yield. On the other hand, most popularly used FYM and vermicompost required more energy input and returned less energy in the system.

**Table 8.** Pearson's correlation among different parameters.

| Parameters | Input Energy | Output Energy | Cost of Cultivation | Net Return | B:C Ratio | Economic Efficiency |
|---|---|---|---|---|---|---|
| Input energy | 1 | | | | | |
| Output energy | 0.340 | 1 | | | | |
| Cost of cultivation | −0.025 | 0.929 ** | 1 | | | |
| Net return | 0.341 | 1.000 ** | 0.929 ** | 1 | | |
| B:C ratio | 0.247 | 0.669 ** | 0.610 * | 0.669 ** | 1 | |
| Economic efficiency | 0.653 ** | 0.917 ** | 0.715 ** | 0.917 ** | 0.648 ** | 1 |

Treatment details are described in Table 1; ** significant at $p < 0.01$; * significant at $p < 0.05$.

**Table 9.** Pearson's correlation among % change from conventional management practices among different parameters.

| Parameters | Cost of Cultivation | Net Return | B:C Ratio | Economic Efficiency | Input Energy | Output Energy |
|---|---|---|---|---|---|---|
| Cost of cultivation | 1 | | | | | |
| Net return | 0.258 | 1 | | | | |
| B:C ratio | −0.094 | 0.935 ** | 1 | | | |
| Economic efficiency | 0.259 | 1.000 ** | 0.935 ** | 1 | | |
| Input energy | 0.298 | 0.716 ** | 0.626 * | 0.715 ** | 1 | |
| Output energy | 0.601 * | 0.910 ** | 0.714 ** | 0.910 ** | 0.708 ** | 1 |

Treatment details are described in Table 1; ** significant at $p < 0.01$; * significant at $p < 0.05$.

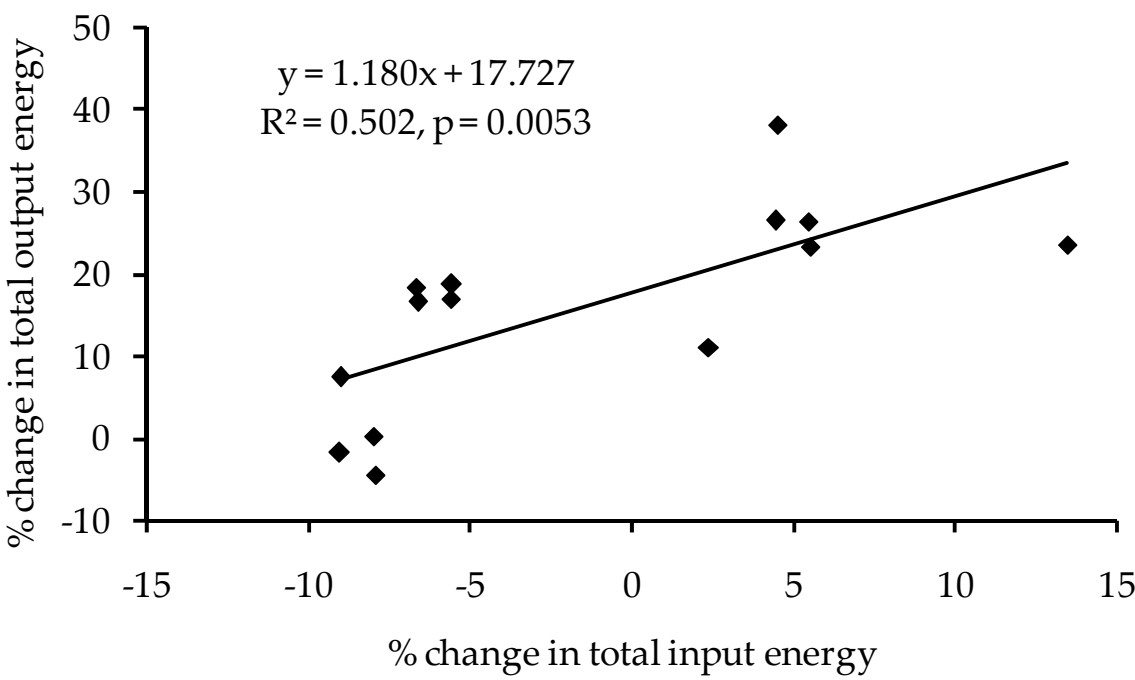

**Figure 1.** The relation between input and output energy in the rice–maize–green gram system.

**Table 10.** Influence of management practices on change in energy input and output over conventional practices.

| Treatment Combinations | Total Energy Input (MJ/ha) | Total Energy Output (MJ/ha) | % Change in Energy Input over Conventional | % Change in Energy Output over Conventional |
|---|---|---|---|---|
| $NM_1 \times WM_1$ | 39,348 | 338,174 | - | - |
| $\times WM_2$ | 40,285 | 375,698 | 2.38 | 11.10 |
| $\times WM_3$ | 44,646 | 417,721 | 13.46 | 23.52 |
| $NM_2 \times WM_1$ | 36,226 | 323,122 | −7.93 | −4.45 |
| $\times WM_2$ | 37,162 | 401,639 | −5.56 | 18.77 |
| $\times WM_3$ | 41,523 | 417,264 | 5.53 | 23.39 |
| $NM_3 \times WM_1$ | 36,206 | 339,194 | −7.99 | 0.30 |
| $\times WM_2$ | 37,142 | 395,687 | −5.61 | 17.01 |
| $\times WM_3$ | 41,503 | 427,757 | 5.48 | 26.49 |
| $NM_4 \times WM_1$ | 35,816 | 363,334 | −8.98 | 7.44 |
| $\times WM_2$ | 36,752 | 394,311 | −6.60 | 16.60 |
| $\times WM_3$ | 41,113 | 467,293 | 4.49 | 38.18 |
| $NM_5 \times WM_1$ | 35,793 | 332,294 | −9.03 | −1.74 |
| $\times WM_2$ | 36,730 | 400,189 | −6.65 | 18.34 |
| $\times WM_3$ | 41,091 | 428,016 | 4.43 | 26.57 |

Treatment details are described in Table 1.

## 4. Conclusions

In this study, it can be concluded that BSM and neem cake could be alternative organic manures in the integrated nutrient management module and they could be able to compensate the paucity of FYM and vermicompost in the country. In this respect, the single application of a pre-emergence herbicide followed by hoeing may be adopted to obtain more yield with less energy expenditure under the rice–maize–green gram system. However, more research information needs to be generated in other cropping systems to confirm the advantages of these sources of nutrients in an integrated nutrient and weed management module. In addition to that, the energy equivalent coefficient of individual organic manures and herbicides specifically needs to be evaluated for better understanding of the process.

**Supplementary Materials:** The following are available online at https://www.mdpi.com/2073-439
5/11/1/166/s1, Table S1: Nutrient content (%) in different organic manures and application rate,
Table S2: Energy calculation for maize production system/hectare, Table S3: Energy utilized through
different nutrient and weed management practices in maize production system/hectare, Table S4:
Energy calculation for green gram production system/hectare, Table S5: Energy utilized through dif-
ferent nutrient (residual) and weed management practices in green gram production system/hectare,
Table S6: Energy calculation for rice production system/hectare, Table S7: Energy utilized through
different nutrient and weed management practices in rice production system/hectare.

**Author Contributions:** Conceptualization, D.G. and K.B.; methodology, D.G. and K.B.; software,
A.D., S.S. and N.K.D.; validation, K.B. and A.H.; formal analysis, D.G., A.D., B.P. and D.M.; inves-
tigation, D.G., A.D., S.S. and N.K.D.; resources, K.B. and A.D.; data curation, D.G., D.M. and B.P.;
writing—original draft preparation, D.G., A.D., P.K.M., S.S. and A.H.; writing—review and editing,
K.B., A.H., S.M. and M.M.H.; supervision, K.B.; project administration, K.B.; funding acquisition,
A.H. and M.M.H. All authors have read and agreed to the published version of the manuscript.

**Funding:** The current work was funded by Taif University Researchers Supporting Project number
(TURSP-2020/59), TaifUniversity, Taif, Saudi Arabia.

**Acknowledgments:** The authors would like to thank Hon'ble Vice-Chancellor Bidhan Chandra
KrishiViswavidyalaya for providing resources to complete the trials. The authors also extend their
appreciation to Taif University for funding current work through the Taif University Researchers
Supporting Project number (TURSP-2020/59), Taif University, Taif, Saudi Arabia.

**Conflicts of Interest:** The authors declare no conflict of interest.

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
