# Peer review of "Assessment of Energy Budgeting and Its Indicator for Sustainable Nutrient and Weed Management in a Rice-Maize-Green Gram Cropping System"

_agronomy, doi:10.3390/agronomy11010166_

Round 1

Reviewer 1 Report

The manuscript “Assessment of energy budgeting and its indicator for sustainable nutrient and weed management in a rice-maize-greengram cropping system” is well written and the methods are in general, technically sound. The experimental treatments used seems to be effective for the proposed analysis.

The authors should consider addressing the following specific comments:

Lines 52-54: This sentence is poorly constructed and doesn’t flow well with the subsequent literature.. . the authors should either revise or consider deleting and starting the sentence with the third sentence … In the Indian….

Lines 65-68: Please provide a citation for the statement

Line 79- Replace the word ‘workers’ with ‘researchers’

Line 83: The statement “among the pest and diseases, weed poses …” This is a poorly constructed statement- one may interpret that weed is a category of pest and diseases. The author should consider revising the statement by replacing… ’pest and diseases’ by ‘crop challenges’

Line 89: use the right English grammar on the sentence …. ‘ because of easy to apply’

Lines 94-95: This statement is not clear .. consider revising

Line 98-99: Incomplete statement

Line 103- replace the word ‘ decide’ by ‘decision’

Line 141: In table 2- change ‘women’ to ‘woman’

Line 211-214: This statements should be supported by a citation.

Line 170- The authors should include the study limitations at the end of this sections

Author Response

Response to Reviewer 1 comments

Comment: Lines 52-54: This sentence is poorly constructed and doesn’t flow well with the subsequent literature.. . the authors should either revise or consider deleting and starting the sentence with the third sentence … In the Indian….

Authors’ response: Deleted as suggested

Comment: Lines 65-68: Please provide a citation for the statement

Authors’ response: Citation added

Comment: Line 79- Replace the word ‘workers’ with ‘researchers’

Authors’ response: Modified as per suggestion

Comment: Line 83: The statement “among the pest and diseases, weed poses …” This is a poorly constructed statement- one may interpret that weed is a category of pest and diseases. The author should consider revising the statement by replacing… ’pest and diseases’ by ‘crop challenges’

Authors’ response: Revised as suggested

Comment: Line 89: use the right English grammar on the sentence …. ‘ because of easy to apply’

Authors’ response: Modified accordingly

Comment: Lines 94-95: This statement is not clear .. consider revising

Authors’ response: Revised, now it is clear

Comment: Line 98-99: Incomplete statement

Authors’ response: Modified

Comment: Line 103- replace the word ‘ decide’ by ‘decision’

Authors’ response: Replaced as per suggestion

Comment: Line 141: In table 2- change ‘women’ to ‘woman’

Authors’ response: Modified as suggested

Comment: Line 211-214: This statements should be supported by a citation

Authors’ response: Citation added

Comment: Line 170- The authors should include the study limitations at the end of this sections

Authors’ response: The limitation of the study has been included at the end.

Reviewer 2 Report

Determining the energy input/output for agriculture production can provide supplementary data to help growers understand the impact of the production practice and how it may be related to yield. Overall, this work summarizes the energy input/output for a specific crop rotation using various nutrient and weed management practices. This is a great concept for a research project, but additional details are required to fully understand how the energy coefficients were developed and which energy components (and quantity) were used for each management practice. My specific comments are in the attached document.

Author Response

Response to Reviewer 2 comments

Comment: Determining the energy input/output for agriculture production can provide supplementary data to help growers understand the impact of the production practice and how it may be related to yield. Overall, this work summarizes the energy input/output for a specific crop rotation using various nutrient and weed management practices. This is a great concept for a research project, but additional details are required to fully understand how the energy coefficients were developed and which energy components (and quantity) were used for each management practice. My specific comments are in the attached document.

Authors’ response: We are very much thankful to the anonymous reviewer for his complement and also suggestion to improve the manuscript.

Comment: State which climate change scenarios you are referring to. What variables and how do they affect crop selection?

Authors’ response: We are referring here the erratic rainfall distribution pattern and fluctuation in temperature (terminal heat wave during February and March). Being a C4 crop, maize can sustain well in the changing climatic scenario due to its photo and thermo insensitiveness.  

Comment: What does less selectivity due to overdose mean? Please explain.

Authors’ response: Even with higher dose of herbicide the selectivity or control efficiency of herbicide may reduce due to development of herbicide resistance in weeds.

Comment: Why did you use different herbicides in the integrated method? Why didn't you use the same herbicides as WM2 then just follow by hoeing?

Authors’ response: The main target was to compare the sole chemical approach with integrated weed management (IWM) approach. In IWM approach, if we apply post-emergence herbicide then mechanical weeding was quite irrelevant due to paucity of time. Thus pre-emergence herbicides were chosen in rice and greengram for IWM treatment. However, in sole chemical approach there was no recommended post-emergence herbicide suitable for target weed flora in maize was available at the time of experimental set up.

Comment: How much total volume/weight of each organic material was applied per ha? How did you calculate %RD of each organic material? What laboratory analysis was used to test %N?

Authors’ response: Supplementary table A has been added for better understanding. Measured amount of organic manure sample was digested with concentrated H2SO4 accelerated with a digestion mixture for 1-2 hours at 420°C until green colour was obtained. The digest was distilled with 40% NaOH and back-titrated with 0.025 (N) H2SO4 by Micro-Kjeldahl’s method (AOAC, 2000).

Comment: Were the organic manures topdressed or incorporated into the soil? If incorporated, what depth?

Authors’ response: Organic manures were incorporated during final land preparation up to 15-20 cm soil depth.

Comment: What is the reference source for these coefficients? You need to cite it. What variables are taken into account when these coefficients were developed? You need to include some summary details in the introduction.

Authors’ response: The references are added as per suggestion. We have obtained the coefficient from secondary sources as mentioned in the Table 2.

Comment: How did all the organic manures have the same coefficient? The are all produced in different manners which would affect input. How was the input coefficient developed and what variables are considered? Also, was the manual/mechanical input required to apply these materials in the field accounted for in your equations? I suggest including a table that defines the amount of each input be listed for each management practice.

Authors’ response: There is no separate energy equivalent coefficients are available for FYM, vermicompost, Brassicaceous seed meal and neem cake viz-a-viz. global literature considered them together as ‘Manure’ which energy equivalent coefficients is 0.3 MJ kg-1. Thus, we are constrained to use the energy equivalent coefficients 0.3 MJ kg-1 for manures. The manual input required for application of organic manures was included in input energy calculation. Supplementary Tables (Table SI to Table S7) were added.

Comment: How does reducing herbicide applications result in improved soil health? You need to provide specific examples in the introduction or delete this statement.

Authors’ response: Modified, as suggested.
